# An Evaluation of Hemostatic Dysregulation in Canine Multicentric Lymphoma

**DOI:** 10.3390/ani14030500

**Published:** 2024-02-02

**Authors:** Maria Ludovica Messina, Fausto Quintavalla, Angelo Pasquale Giannuzzi, Tommaso Furlanello, Marco Caldin

**Affiliations:** 1Division of Internal Medicine, Department of Veterinary Sciences, University of Parma, 43121 Parma, Italy; 2Division of Internal Medicine, San Marco Veterinary Clinic, 35030 Veggiano, Italy

**Keywords:** dog, lymphoma, hemostatic parameters, hypercoagulability, fibrinogen, platelet

## Abstract

**Simple Summary:**

Canine lymphoma is the most common malignant hematopoietic tumor. It is mainly seen in middle-aged or older dogs, and it is characterized by a clonal proliferation of B or T lymphocytes. The development of hemostatic changes is associated with malignant tumors in human medicine and is also reported in dogs. These hemostatic alterations may contribute unfavorably to the morbidity and mortality associated with the disease. The aim of this study was to evaluate, in a large group of dogs affected by lymphoma compared to dogs with other diseases, whether there is an alteration of hemostatic parameters in canine lymphoma and, if so, whether these were correlated with the type (B or T) of lymphoma and with the stage of the disease.

**Abstract:**

Multiple hemostatic abnormalities are associated with paraneoplastic syndrome and some malignant tumors. Lymphoma is the most common hematopoietic neoplasm in dogs, sometimes associated with hemostatic changes. The objectives of this study were to evaluate the behavior of coagulation parameters in dogs with multicentric lymphoma compared with diseased dogs without lymphoma, to separately evaluate the effect of immunophenotype (B lymphoma versus T lymphoma) on the variables of interest as well as the effect of disease stage (stage II to IV versus stage V). Specifically, a cross-sectional study was performed with a matched comparison group considering 170 dogs with B or T lymphoma (group 1) and 170 dogs with no lymphoma or other neoplastic processes but other diseases (group 0). Eight coagulation parameters were evaluated: platelet count (Plt), activated partial thromboplastin time (aPTT), prothrombin time (PT), thrombin time (TT), fibrinogen, fibrin/products of fibrinogen degradation (FDPs), fibrin D-dimers, and antithrombin (AT). Dogs with lymphoma showed prolonged PT and TT, decreased fibrinogen, increased FDP, and decreased Plt compared with group 0. The effect of disease stage was evaluated separately for dogs with stage II to IV lymphoma and dogs with stage V lymphoma; patients with stage II–IV lymphoma showed no significant differences, while in dogs with stage V lymphoma, a prolongation of PT and TT, a decrease in fibrinogen, an increase in FDPs and a decrease in Plt were found compared with the group 0. Finally, the comparison between B lymphoma and T lymphoma showed no significant differences in coagulation parameters between the two groups. Logistic regression analysis demonstrated that low fibrinogen and platelet levels were the most significant predictors of lymphoma in a cohort of canine patients. These hemostatic abnormalities in lymphoma appeared to be associated with the stage of the disease rather than the lymphoma immunophenotype. These findings pave the way for the possible scenario of lymphoma-associated fibrinolysis and the so far undescribed pattern of hyperfibrinolysis associated with the most severe stage of lymphoma.

## 1. Introduction

Canine lymphoma is a malignant tumor characterized by a clonal proliferation of B or T lymphocytes; this proliferation usually takes place in solid tissues (such as the lymph nodes, spleen, liver, etc.), but it can develop in almost any tissue of the canine body [1,2]. It follows that the symptoms of lymphoma are not specific and are very variable.

Canine lymphoma is the most common hematopoietic tumor in dogs, and it is mainly seen in middle-aged or older dogs (average age is between 6 and 9 years), even if T-cell lymphoma tends to affect younger patients [3]. It can affect any dog breed, but middle-sized to larger dogs are the most commonly represented [4]. Regarding clinical presentation, molecular biology, treatment and treatment response, there are many common points between canine lymphoma and human non-Hodgkin lymphoma (NHL), so much so that it is considered as a possible pathogenetic model [1,5,6,7,8]. Most dogs (>80%) are diagnosed with the multicentric (generalized) form, and specifically, from a histological standpoint, with diffuse large B-cell lymphoma [5,9]. The etiology is multifactorial and includes viral infections, genetic predisposition, and environmental factors, that can be influenced by an imbalance of the redox status [6,10].

The development of hemostatic changes is one of the most common paraneoplastic syndromes in dogs [11,12], and these disorders can range from hemorrhages to thrombosis [13,14,15,16,17,18,19]. Hemostatic changes associated with malignant tumors are reported in human medicine, and they generally cause a hypercoagulable state in the patient which can lead to thrombosis and thromboembolism [20,21,22]. Tumor-associated coagulopathy is also reported in dogs (microthrombi have been documented in dog tumor tissues), and its pathogenesis is still considered complex [23,24,25,26]. In particular, it was observed in dogs with adenocarcinoma, non-mammary carcinomas, hemangiosarcomas, mastocytoma, osteosarcoma, and dogs with distant metastatic disease. Canine lymphoma has not only been associated with thrombocytopenia but also with hypercoagulability and subsequent thromboembolism [27,28]. These hemostatic alterations may contribute to morbidity and mortality associated with the disease [5,29,30,31,32,33,34]. Moreover, hypercoagulability in dogs with lymphoma may be further exacerbated by chemotherapy treatments, such as glucocorticoid and L-asparaginase [28,35,36]. However, the incidence and prognostic significance of thrombosis and/or thromboembolism in dogs with lymphoma are still little studied [28,33,37,38].

The aim of this retrospective analysis was to evaluate whether there is an alteration of hemostatic parameters in canine multicentric lymphoma and, if so, whether these were correlated with the type (B or T) of lymphoma and with the stage of the disease (stage II to V).

## 2. Materials and Methods

### 2.1. Animals and Method

A cross-sectional study with a matched comparison group was performed considering 170 dogs affected by B or T multicentric lymphoma (affected or group 1) and 170 dogs not affected by lymphoma or other neoplastic processes but by other diseases (unaffected or group 0). Specifically, dogs with heart disease, respiratory, gastrointestinal, and skin diseases were enrolled in Group 0. The study was carried out in the period between 1 January 2016 and 31 December 2022 on dogs undergoing clinical examinations at the San Marco Veterinary Clinic, Veggiano (PD), Italy. Dogs were selected using the electronic database, P.O.A System Plus 9.0^®^ (POA System, Venice, Italy). Diagnosis of lymphoma was made using the WHO scheme and directions of the European canine lymphoma networks [39,40,41]. All dogs with lymphoma (group 1) underwent cytopathological and histopathological evaluation of a lymph node, as well as flow cytometry, bone marrow sampling and abdominal ultrasound. An individual matching technique with the affected dogs was used, identifying subjects belonging to the same breed, sex, reproductive status and age (±6 months). If it was not possible to identify an unaffected dog of the same breed as an affected individual, mixed breed dogs of the same sex, reproductive status, age, and weight (±5 kg) were included to complete the matchings. In the event that two or more dogs met the same criteria in an overlapping manner, selection was performed randomly by the software. Patients in both groups had a complete medical record and hematological, biochemical, electrophoretic, urinary and coagulation evaluations.

Exclusion criteria applied to both groups included therapeutic treatments capable of altering the hemostatic process (blood/plasma transfusion, heparin, acetylsalicylic acid, clopidogrel, glucocorticoids, and antineoplastic chemotherapy).

### 2.2. Blood Collection and Sample Analysis 

Before proceeding to establish a standard chemotherapy protocol or any other therapeutic approach, after shaving and disinfecting the sampling area from all the enrolled animals, a blood sample was taken from the cephalic vein (medium or large dogs) or jugular vein (small dogs), using devices attachable to Vacutainer^®^ tubes (Becton Dickinson, Milan, Italy). The blood was then dispensed directly into tubes containing 3.2% sodium citrate as an anticoagulant (Greiner Bio-One^®^ Kremsmünster, Austria). Cell Blood Count (CBC) was performed using the Siemens Healthcare Diagnostic ADVIA 120 and the 2120 hematology analyzer (Siemens Healthinners, Milan, Italy); coagulation testing was performed using the Stago STA-R coagulation analyzer (Stago, Asniéres sur Seine, France), as previously described [42]. Phenotypic diagnosis of B or T lymphoma was performed using the Cytomics FC 500 Beckman Coulter cytofluorimeter (Beckman Coulter, Brea, IN, USA).

### 2.3. Aims and Comparison Groups

Eight coagulation parameters were evaluated: -platelet count (Plt)-prothrombin time (PT)-activated partial thromboplastin time (aPTT)-thrombin time (TT)-fibrinogen-fibrin/fibrinogen degradation products (FDPs)-fibrin D-dimers-antithrombin (AT).

The objectives of this study were to evaluate the following: (a) the behavior of coagulation parameters in dogs with multicentric lymphoma compared with sick dogs without lymphoma; (b) the effect separately of the phenotypes (B lymphoma versus T lymphoma) on the variables of interest; (c) the effect of the stage of the disease (stage II to IV versus stage V) with respect to the study parameters.

In order to achieve the stated objectives, a comprehensive comparison was conducted among dogs with lymphoma, encompassing overall and phenotype-based subgroups. To address potential confounding effects stemming from disease stage, comparisons between B-cell lymphoma and T-cell lymphoma cases were undertaken before and after balancing the stages within each group. Random sampling was employed to ensure a similar number of cases across all lymphoma stages (Table 1), thereby homogenizing the groups. Subsequently, three further comparisons were carried out to investigate the impact of disease stage on study parameters: (1) patients with lymphoma stages II to IV versus the control group, (2) patients with lymphoma stage V versus the control group, and (3) a comparison of stage II/IV with stage V. The relevant variables were assessed utilizing multivariate analysis to identify suitable predictors.

### 2.4. Statistical Analysis

The sample size of each group was estimated based on the following assumptions: a 1:1 ratio between the affected dogs and the matched control group; the expected effect size expressed via Cohen’s d index is equal to 0.5 (medium effect); type I error was set to 5% and type II error was set to 20%. 

Based on this calculation the number of animals needed in each group and for each comparison was at least 34. Data were studied by assessing the normality distribution (Shapiro-Wilk test). Normally distributed data were reported as the mean and standard deviation (SD) and were evaluated via the T-test for unpaired samples. Not normally distributed data were reported as the median and interquartile range (IQR) and were evaluated using the Wilcoxon-Mann-Whitney test for unpaired samples (Wilcoxon sum rank test). Logistic regression models were also created, for every comparison. 

## 3. Results

Of 170 dogs with multicentric lymphoma, 91 dogs were affected by B-cell lymphoma, and 79 dogs were affected by T-cell lymphoma. Dogs affected by lymphoma from stage II to IV were 62, and patients with stage V counted as 108. Subgroups distinguished by immunophenotype and staging are summarized in Table 2. The control group was matched for sex and sexual status to the affected group with 89 males (70 sexually intact and 19 neutered) and 81 females (18 sexually intact and 63 spayed). In the control group, the mean age was 8.66 ± 3.41 years (median 99; IQR = 9 to 207), and the mean weight was 23.7 ± 14.2 (median 23.6; IQR = 1.7 to 60.0) kg. In the affected group, the mean age was 8.66 ± 3.41 years (median 99.5; IQR = 9 to 208), and the mean weight was 24.7 ± 14.8 (median 24.5; IQR = 1.3 to 69.0) kg. The breed matching of the control group versus the lymphoma group was 95.2% (162 out of 170).

In consideration of the non-parametric distribution of all study variables (Shapiro–Wilk normality test: <0.001), differences between the two groups were analyzed through a Wilcoxon sum rank test for independent samples. A comparative analysis between dogs with lymphoma and sick dogs without lymphoma revealed notable hemostatic abnormalities, including prolonged prothrombin time (PT), decreased fibrinogen, increased FDPs levels, reduced thrombin time (TT), and decreased platelet counts (Table 3). Logistic regression analysis identified fibrinogen and platelet count as the strongest predictors of lymphoma (Table 4). More specifically, higher fibrinogen levels and greater platelet counts were associated with a lower likelihood of lymphoma compared to that of other diseases in dogs (Figure 1 and Figure 2).

To evaluate the influence of immunophenotype on coagulation abnormalities, comparative analysis was performed between B-cell and T-cell lymphoma cases, both before and after equalizing the distribution of disease stages within each group. Initial comparisons without stage balancing revealed that dogs with T-cell lymphoma exhibited prolonged prothrombin time (PT) and significantly decreased platelet counts. Considering these results, it is crucial to recognize that, as presented in Table 2, stage V Lymphoma was overrepresented in the T-cell group within our population. Subsequent comparisons after stage balancing did not reveal statistically significant differences between the two groups, indicating that the observed disparities were likely impacted by the confounding effect of disease stage.

A comparison between dogs with lymphoma stages II to IV and those with stage V lymphoma revealed that the severity of the disease significantly impacts hemostatic test alterations (Table 5). Specifically, dogs with stage V lymphoma exhibited substantially higher levels of FDPs and were more prone to thrombocytopenia compared to dogs with stage II to IV lymphoma. Additionally, they demonstrated prolonged PT and reduced antithrombin activity (Table 5). Box plots demonstrate the impact of lymphoma stage on platelet levels within the immunophenotypes. Dogs with stage V lymphoma exhibited the lowest median platelet count compared to those with stage II–IV lymphoma (Figure 3).

These findings were further corroborated by separate comparisons between the different lymphoma stages and their corresponding control groups. In this case, dogs with stage V lymphoma also displayed elevated levels of D-dimers (Table 5).

On the other hand, the hemostatic alterations observed between dogs with stage V lymphoma and those with stage II to IV lymphoma did not reach statistical significance in the logistic regression model (Table 6).

Finally, logistic regression models performed by analyzing each lymphoma stage separately relative to its corresponding control group yielded results consistent with the overall lymphoma group comparison: fibrinogen and platelet count emerged as significant predictors of the outcome (Table 6).

## 4. Discussion

In vivo hemostasis is a highly dynamic process [28,43]. All neoplastic and non-neoplastic diseases can influence the coagulation system and its derangement is a common phenomenon in critically ill patients [44]. Since any inflammatory or disease state can influence coagulation, we investigated whether a standard coagulation profile encompassing platelet count would exhibit variations in a large cohort of dogs with multicentric lymphoma. The comparison was conducted against dogs with cardiac, respiratory, gastrointestinal, or dermatologic diseases, excluding those with purely inflammatory conditions or critical illnesses. To minimize bias arising from age, breed, sex, and sexual dimorphism, control dogs were meticulously selected to match the characteristics of dogs with lymphoma.

Table 3 shows that several blood clotting parameters, including prothrombin time (PT), thrombin time (TT), fibrinogen levels, fibrin degradation products (FDPs), and platelet count, were significantly different in dogs with lymphoma compared to sick dogs without lymphoma. This pattern was observed regardless of the lymphoma subtype (B cell or T cell). Although T cell lymphoma is generally considered to have a poorer prognosis than B cell lymphoma, our analysis of the coagulation profile did not reveal any significant differences between the two subtypes. However, when we compared dogs with stage V lymphoma to those with lower stages, we found that the severity of the lymphoma, rather than the lymphoma subtype, significantly affected the coagulation system. This was clearly observed in Table 4, where the majority were very similar to what obtained in the control group, while significant differences emerged matching stage V lymphoma against the control group. Although alterations in the coagulation profile are not widely considered as a reliable prognostic indicator in canine lymphoma [36,37,45], the association with the most severe stage of the disease warrants further consideration.

The intricate relationship between hemostasis and malignancy has long been recognized, with studies in both humans and dogs demonstrating the influence of coagulation factors on tumor biology [43,44]. These factors play a direct role in promoting tumor growth, aggressiveness, and metastatic potential [7,18,46,47,48,49]. In human oncology, thrombotic complications, including deep vein thrombosis, pulmonary embolism, and disseminated intravascular coagulation (DIC), are common and frequently fatal side effects of malignancy, including lymphoma [11,50,51,52,53]. For canine lymphoma, low platelet count, hypercoagulability, and hemostatic disorders have been documented in affected dogs and have been associated with poor prognosis [28,48]. However, these previous studies did not comprehensively evaluate the coagulation parameters covered in our current investigation.

PT and TT were elongated in dogs affected by lymphoma, while fibrinogen was lower. This pattern, to our knowledge, never described in canine cases, could be associated with a tendency to hypercoagulability, as described by Kol and coworkers [28]. This should be associated with an increase in D-dimers, which was not observed in our case, while we measured increased FDPs. In general, the situation observed in our study may be related to consumption due to thrombosis or disseminated intravascular coagulation (DIC), although the sensitivity of D-dimers for DIC is still a matter of debate [54]. DIC is an acquired syndrome characterized by a severe complication of life-threatening disorders. The main pathophysiological mechanisms of DIC that develop are a consequence of crosstalk between the coagulation and inflammatory host-defense systems and secondary imbalances in pro- and anticoagulant factors. Dogs with tumors usually exhibited elevated levels of D-dimers [55,56]. As reported in our study, no statistically significant variations in D-dimer concentration were found between dogs with lymphoma and control dogs, nor between the various stages of the neoplastic disease, although the values tend to increase with the stage of the disease. 

Our study uncovered an intriguing coagulation profile in dogs with lymphoma: prolonged prothrombin time (PT) and thrombin time (TT) were accompanied by reduced fibrinogen levels compared to dogs affected by other diverse conditions. This finding, previously undocumented in canine lymphoma cases, suggests the possibility of a hypercoagulable state, as posited by Kol et al. [28]. However, contrary to this hypothesis, no significant increase in D-dimers was observed, while FDPs were indeed elevated. This seemingly paradoxical finding could nevertheless be attributed to the consumption of fibrinogen due to thrombosis or disseminated intravascular coagulation (DIC), given the ongoing debate regarding the sensitivity of D-dimers as a DIC marker [54]. DIC is a severe complication of life-threatening disorders characterized by a dysregulation of the coagulation system. The underlying pathophysiological mechanisms involve crosstalk between coagulation and inflammatory pathways, leading to imbalances in pro- and anticoagulant factors. Elevated D-dimers are commonly reported in dogs with tumors [55,56]. In contrast to a previous study [8], our investigation did not reveal statistically significant differences in D-dimer levels between dogs with lymphoma and control dogs, or between dogs with different lymphoma stages.

Another explanation for the lower fibrinogen levels in dogs with stage V lymphoma is pathological fibrinolysis, a condition characterized by excessive plasmin activation that occurs independently of the normal coagulation cascade. When this uncontrolled plasmin activity overwhelms the neutralizing effects of its inhibitors, it can lead to severe bleeding complications. In a 2022 study, Zoia et al. demonstrated that coagulation followed by fibrinogenolytic activity occurs in dogs in all types of ascitic and pleural fluids, and that this increased fibrinolytic activity is also present in their blood, in some cases reaching a hyperfibrinolytic state [57]. While there is limited evidence of fibrinolysis in hematological malignancies [58,59], hyperfibrinogenolysis has also been detected in dogs with angiostrongylosis, neoplasia, liver disease, traumatic hemorrhagic shock, and other diseases [45,57]. Based on the findings of the present study, a scenario of primary lymphoma-associated hyperfibrinolysis can be hypothesized. Furthermore, having detected coagulation changes only in patients with stage V lymphoma, hyperfibrinogenolysis would appear to be associated with the more severe stages of lymphoma, potentially serving as a marker of disease severity.

A recent review underscored the importance of thrombocytopenia as a predictor for both remission and survival in lymphoma [45], even surpassing other prognostic factors in a study involving 98 dogs with diffuse large B-cell lymphoma, following a chemotherapy [48]. The underlying mechanisms of thrombocytopenia associated with neoplasia encompass a combination of factors, including augmented platelet consumption, decreased platelet production secondary to myelodysplasia, immune-mediated platelet destruction, and sequestration of platelets [16,17,46,48,60,61]. In human literature, immune-mediated platelet destruction is frequently implicated in lymphoma-related thrombocytopenia [62]. However, infiltration of the bone marrow, the hallmark of stage V lymphoma, is also likely to contribute to this finding. Our study revealed a correlation between increasing lymphoma stage and decreasing platelet count, although the smaller sample size for stage IV compared to stage V could influence this observation.

Another likely explanation for thrombocytopenia in lymphoma patients, consistent with findings in both human and veterinary medicine, is the development of lymphoma-associated hemophagocytic syndrome (LAHS). Hemophagocytic syndrome (HPS) is a hyperinflammatory state characterized by dysregulation of the immune system, leading to the excessive secretion of proinflammatory cytokines. These cytokines contribute to the development of multiple cytopenias, including thrombocytopenia [63]. Hypofibrinogenemia, commonly observed in our cases, is a well-established feature of human LHAS [64]. Acquired HPS can be associated with neoplasia, and lymphoma is the most common neoplasm associated with this condition [65]. Recent reports have documented episodes of LAHS and thrombocytopenia in a small number of dogs [63,64]. Notably, the vast majority of cases, both in human and canine populations, are associated with T-cell lymphoma. An intriguing finding arising from our study is that the lowest platelet counts belonged to dogs affected by the same phenotype.

## 5. Conclusions

Dogs frequently develop high-grade aggressive multicentric lymphoma, comparable to human non-Hodgkin lymphoma. The present investigation describes, for the first time, significant abnormalities in primary, and secondary coagulation and fibrinolysis in dogs affected by multicentric lymphoma in comparison to other medical conditions. Interestingly, those differences are not truly related to the presence of the lymphoma, but to its severity and dogs classified as stage V, irrespective of the immunophenotype, display laboratory signs of possible hyperfibrinolysis. The percentage of cases in stage V observed in the authors’ practice, was abundant enough to create a statistical difference in the coagulation profile of dogs affected by lymphoma against non-neoplastic dogs. Logistic regression analysis identified fibrinogen and platelet count as the strongest predictors of lymphoma.

These findings pave the way for the possible scenario of lymphoma-associated fibrinolysis, and the so far undescribed pattern of hyperfibrinolysis associated with the most severe stage of lymphoma. This would have further importance on the choice of the type of chemotherapy regimen to be implemented or establish adequate prophylaxis before undertaking a therapeutic approach.

Further studies are needed to better characterize this phenomenon and the potential clinical implications of its use as a marker of severity of the lymphomatous disease.

## Figures and Tables

**Figure 1 animals-14-00500-f001:**
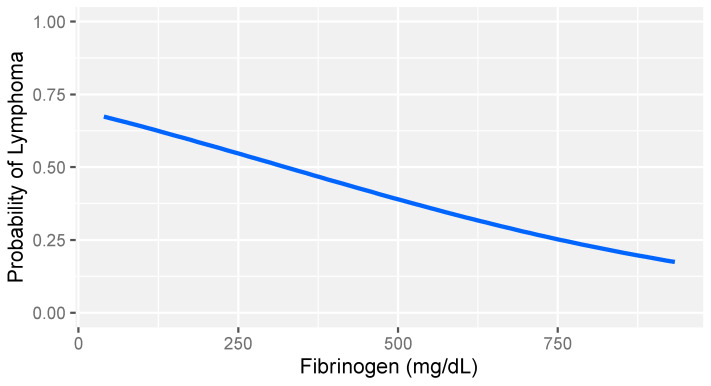
Logistic regression model: probability of lymphoma versus that of other diseases based on fibrinogen levels in dogs with stage V lymphoma. The gray band represents the 95% confidence interval.

**Figure 2 animals-14-00500-f002:**
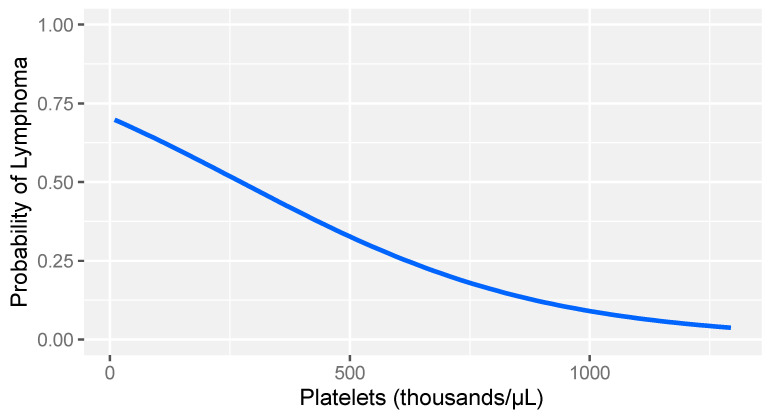
Logistic regression model: probability of lymphoma versus other diseases based on platelet count levels in dogs with stage V lymphoma. The gray band represents the 95% confidence interval.

**Figure 3 animals-14-00500-f003:**
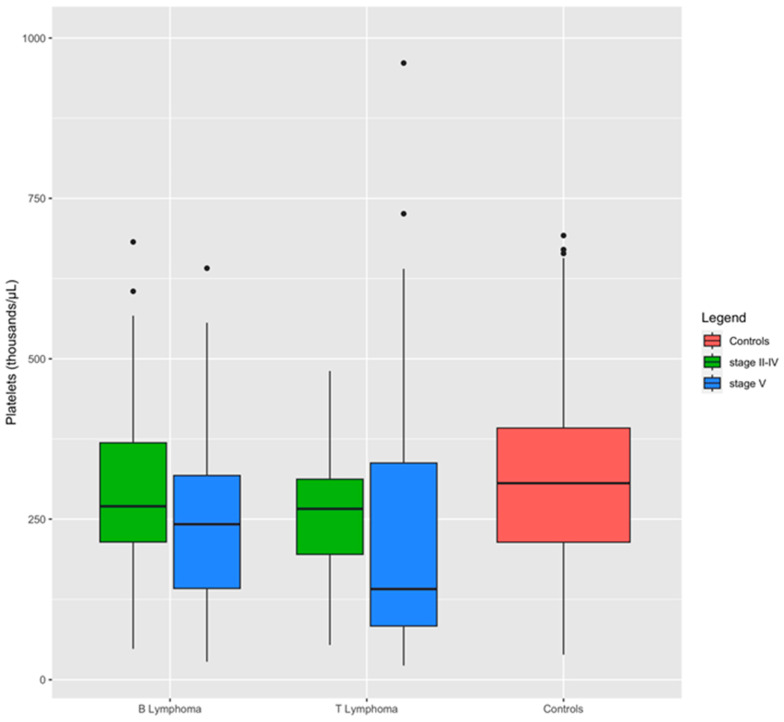
Distribution of platelets between groups and lymphoma subgroups. Boxplots show the influence of stage on platelet level within the immunophenotypes: dogs affected by stage V of lymphoma have a median level of platelets lowest than dogs with stage II to IV of the disease.

**Table 1 animals-14-00500-t001:** B lymphoma group and T lymphoma group balanced according to stage.

	Lymphomas B N = 56	Lymphomas T N = 56
**Stage**		
stage III	6 (11%)	6 (11%)
stage IV	7 (12%)	7 (12%)
stage V	43 (77%)	43 (77%)

**Table 2 animals-14-00500-t002:** Lymphoma group: distinction into subgroups by immunophenotype and stage.

Stage	Lymphomas B N = 91	Lymphomas T N = 79
stage II	0 (0%)	1 (1.3%)
stage III	6 (6.6%)	6 (7.6%)
stage IV	42 (46%)	7 (8.9%)
stage V	43 (47%)	65 (82%)

**Table 3 animals-14-00500-t003:** Analysis of the association between phenotypes and hemostatic changes: comparison between groups.

Variables (Units)	RI	Lymphoma (N = 170) *vs*. Control (N = 170)			B Lymphoma (N = 91) *vs*. T Lymphoma (N = 79) (without Balancing the Stages)			B Lymphoma (N = 56) *vs*. T Lymphoma (N = 56) (after Balancing the Stages)		
		Lymphoma Median (IQR)	Control Group Median (IQR)	*p*	B Lymphoma Median (IQR)	T Lymphoma Median (IQR)	*p*	B Lymphoma Median (IQR)	T Lymphoma Median (IQR)	*p*
**PT** **(s)**	6.9–8.6	8.10 (7.70, 8.80)	8.00 (7.53, 8.50)	**0.005**	8.00 (7.70, 8.55)	8.30 (7.75, 9.30)	**0.032**	8.05 (7.70, 8.70)	8.05 (7.70, 9.60)	0.3
**aPTT** (s)	10.1–12.5	12.20 (11.60, 12.80)	12.40 (11.80, 13.18)	0.056	12.20 (11.60, 12.55)	12.3 (11.5, 13.0)	0.3	12.2 (11.7, 12.7)	12.3 (11.8, 13.1)	0.4
**Thrombin Time** (s)	11.6–15.2	13.40 (12.30, 14.98)	13.00 (12.00, 14.18)	**0.022**	12.90 (12.10, 14.70)	13.7 (12.6, 15.2)	**0.024**	13.6 (12.3, 15.0)	14.3 (12.7, 15.6)	0.2
**Fibrinogen** (mg/dL)	131–241	259.5 (183.0, 385.2)	308.5 (216.2, 475.2)	**<0.001**	260.0 (199.5, 399.0)	259.0 (160.5, 370.5)	0.2	250 (189, 429)	234 (167, 339)	0.2
**D-Dimers** (µg/mL)	0.01–0.5	0.09 (0.04, 0.17)	0.08 (0.04, 0.14)	0.2	0.08 (0.04, 0.16)	0.11 (0.06, 0.18)	0.12	0.10 (0.04, 0.18)	0.12 (0.07, 0.18)	0.3
**FDP_s_** (µg/mL)	011–1.97	2.89 (1.23, 6.50)	1.48 (0.75, 3.62)	**<0.001**	3.38 (1.32, 9.47)	2.36 (1.12, 5.56)	0.2	4 (2, 16)	3 (1, 6)	0.087
**Antithrombin** (%)	108–155	116.0 (102.2, 130.7)	115.5 (103.0, 129.5)	0.8	117.00 (103.50, 130.50)	115.0 (101.5, 130.5)	0.7	116 (98, 126)	121 (101, 137)	0.3
**Platelets** (thousands/µL)	180–451	237.00 (108.0, 336.5)	303.5 (213.2, 391.0)	**<0.001**	252.0 (166.0, 349.5)	168.0 (85.0, 334.5)	**0.011**	243 (145, 331)	194 (84, 364)	0.3

RI: Reference Intervals *p*: *p*-value (Wilcoxon rank sum test). Statistically significant values are in bold.

**Table 4 animals-14-00500-t004:** Analysis of the association between phenotypes and hemostatic changes: logistic regression models. TT was excluded from regression models to avoid multicollinearity as it is strongly correlated with fibrinogen levels.

Variables (Units)	Lymphoma *vs*. Control				B Lymphoma *vs*. T Lymphoma (without Balancing the Stages)				B Lymphoma *vs*. T Lymphoma (after Balancing the Stages)			
	β	OR	IC	*p*	β	OR	IC	*p*	β	OR	IC	*p*
**PT** (s)	0.005	1	0.93–1.07	0.8	0.30	1.36	1.03–1.9	**0.049**	0.24	1.27	0.96–1.8	0.14
**aPTT** (s)	−0.005	0.99	0.90–1.07	0.9	0.01	1.01	0.9–1.2	0.8	0.02	1.03	0.88–1.2	0.7
**Fibrinogen** (mg/dL)	0.0019	0.998	0.996–0.999	**0.009**	−0.002	0.99	0.9–1	0.8	0.001	0.99	0.99–1.0	0.3
**D-Dimers** (µg/mL)	−0.092	0.91	0.73–1.09	0.3	0.49	1.63	0.9–3.2	0.09	0.53	1.69	0.91–4.19	0.14
**FDP_s_** (µg/mL)	−0.002	0.99	0.98–1.01	0.9	−0.02	0.98	0.9–1	0.06	−0.18	0.98	0.95–1	0.1
**Antithrombin** (%)	0.003	1	0.99–1.01	0.5	0.001	1	0.9–1	0.8	0.005	1	0.99–1	0.5
**Platelets** (thousands/µL)	−0.002	0.997	0.996–0.999	**0.006**	−0.0008	0.99	0.9–1	0.4	0.0004	1	0.99–1	0.7

β: coefficients; OR: odds ratio; IC: confidence interval of OR; *p*: *p*-value. Statistically significant values are in bold.

**Table 5 animals-14-00500-t005:** Analysis of the association between stages of lymphoma and hemostatic changes: comparison between groups.

Variables (Units)	RI	II–IV Stage Lymphoma (N = 62) *vs*. Control Group (N = 62)			V Stage Lymphoma (N = 108) *vs*. Control Group (N = 108)			II–IV Stage Lymphoma (N = 62) *vs*. V Stage Lymphoma (108)		
		II–IV Stage Lymphoma Median (IQR)	Control Group Median (IQR)	*p*	V Stage Lymphoma Median (IQR)	Control Group Median (IQR)	*p*	II–IV Stage Lymphoma (N = 62), Median (IQR)	V Stage Lymphoma (N = 108), Median (IQR)	*p*
**PT** **(s)**	6.9–8.6	7.90 (7.7, 8.4)	8.00 (7.53, 8.50)	>0.9	8.30 (7.70, 9.30)	7.90 (7.58, 8.60)	**<0.001**	7.90 (7.70, 8.40)	8.30 (7.70, 9.30)	**0.006**
**aPTT** (s)	10.1–12.5	12.1 (11.5, 12.4)	12.5 (11.7, 13.4)	**0.009**	12.30 (11.70, 13.00)	12.3 (11.8, 13.0)	0.7	12.1 (11.5, 12.4)	12.3 (11.7, 13.0)	0.064
**Thrombin Time** (s)	11.6–15.2	13.1 (12.1, 14.9)	12.8 (12.0, 14.0)	0.2	13.50 (12.38, 15.00)	13.1 (12.0, 14.2)	0.058	13.10 (12.10, 14.90)	13.5 (12.3, 15.0)	0.5
**Fibrinogen** (mg/dL)	131–241	255 (207, 379)	293 (208, 441)	0.3	260 (161, 387)	330 (228, 488)	**<0.001**	255.00 (207.00, 379.25)	259.5 (160.7, 386.5)	0.3
**D-Dimers** (µg/mL)	0.01–0.5	0.08 (0.03, 0.14)	0.09 (0.03, 0.14)	0.6	0.11 (0.04, 0.18)	0.08 (0.05, 0.14)	**0.040**	0.08 (0.03, 0.14)	0.11 (0.04, 0.18)	0.066
**FDP_s_** (µg/mL)	011–1.97	2 (1, 4)	1 (0, 3)	**0.021**	4 (1, 8)	2 (1, 4)	**<0.001**	2.10 (1.07, 4.42)	3.67 (1.32, 8.24)	**0.021**
**Antithrombin** (%)	108–155	122 (107, 138)	120 (105, 133)	0.3	114 (99, 126)	114 (99, 128)	0.7	122.00 (107.25, 137.75)	114.0 (99.0, 126.2)	**0.014**
**Platelets** (thousands/µL)	180–451	270 (207, 369)	294 (200, 361)	0.7	173 (86, 331)	309 (221, 407)	**<0.001**	270.0 (206.7, 368.5)	172.5 (85.5, 331.0)	**0.001**

RI: Reference intervals *p*: *p*-value (Wilcoxon rank sum test). Statistically significant values are in bold.

**Table 6 animals-14-00500-t006:** Analysis of the association between stages of lymphoma and hemostatic changes: logistic regression models. TT was excluded from regression models to avoid multicollinearity as it is strongly correlated with fibrinogen levels.

Variables (Units)	Stage II–IV Lymphoma *vs*. Control Group				Stage V Lymphoma *vs*. Control Group				Stage II–IV Lymphoma *vs*. Stage V Lymphoma			
	β	OR	IC	*p*	β	OR	IC	*p*	β	OR	IC	*p*
**PT** (s)	0.005	1	0.93–1.07	0.8	0.0008	0.99	0.91–1.07	0.9	0.46	1.59	0.001–2.6	0.05
**aPTT** (s)	−0.005	0.99	0.90–1.07	0.9	0.01	1	0.9–1.14	0.8	0.06	1.06	0.18–1.3	0.5
**Fibrinogen** (mg/dL)	0.0019	0.998	0.996–0.999	**0.009**	−0.001	0.998	0.996–0.999	**0.02**	0.001	1	0.019–1	0.3
**D-Dimers** (µg/mL)	−0.092	0.91	0.73–1.09	0.3	−0.14	0.87	0.6–1.05	0.16	0.28	1.32	0.017–2.6	0.3
**FDP_s_** (µg/mL)	−0.002	0.99	0.98–1.01	0.9	0.008	1	0.99–1.02	0.3	0.006	1	0.019–1.03	0.7
**Antithrombin** (%)	0.003	1	0.99–1.01	0.5	0.002	1	0.99–1.01	0.7	−0.005	0.99	0.019–1	0.4
**Platelets** (thousands/µL)	−0.002	0.997	0.996–0.999	**0.006**	−0.002	0.997	0.995–0.999	**0.005**	−0.004	0.99	0.019–1	0.7

β: coefficients; OR: Odds ratio; IC: Confidence interval of OR; *p*: *p*-value. Statistically significant values are in bold.

## Data Availability

The data presented in this study are available upon request from the corresponding author.

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
