# Peer review of "An Evaluation of Hemostatic Dysregulation in Canine Multicentric Lymphoma"

_animals, 2024, doi:10.3390/ani14030500_

Round 1

Reviewer 1 Report

Comments and Suggestions for Authors

Thank you for your interesting research about this topic. Indeed, I was surprised by the fact that there were higher number of dogs in stage V lymphoma.This raised a question about the importance to perform a fully staging in these patients.

Matherial and methods:
- Line 70: An unaffected 

- The authors should consider adding the interval reference for the hemostatic parameters in tables.

Discussion:

- Consider to add these references and discuss between their results and yours.

https://pubmed.ncbi.nlm.nih.gov/23710569/

https://actavetscand.biomedcentral.com/articles/10.1186/1751-0147-54-3

Comments on the Quality of English Language

In my opinion, the work has a good English quality.

Author Response

We want to thank you because with your valuable suggestions and comments we hope to have improved our article.

We have provided to insert the interval reference for the hemostatic parameters in tables and we have included the recommended references.

12.20.2023

Based on the indications provided by the editor regarding the length of the text, we have made the necessary corrections. It follows that some parts of the text and in particular the “Discussion”, highlighted in yellow, have undergone a remodelling. We apologize for the confusion caused.

FQ

Reviewer 2 Report

Comments and Suggestions for Authors

Thank you for your hard work on this manuscript.  I have some specific edits below but in general, one thing that I think is important to evaluate is the question of whether these results are clinically relevant.  I recognize that they are statistically different but an fdp difference of 2 (1, 4) in the control group versus 4 (1, 8) in the neoplastic group is not particularly of clinical relevance to me in general. The differences presented in this paper are quite mild except for the thrombocytopenia values in the control versus stage 5 lymphoma patients.  So it's just something to consider to point out potentially in that to me, not many of these differences are particularly of clinical relevance but are mainly statistically different.  

I do not think we have solid evidence that there was clinically relevant hyperfibrinolysis occurring in these dogs but I think the conclusions are soft enough that it's still fair to theorize/hypothesize.  

There are also no results for the bone marrow cytology so that would greatly affect platelet count so those results should be included. Also, there was a mention of electrophoretic evaluation which also wasn't elaborated on. If the electrophoresis and urinary parameters are not necessary to the authors to report, then I would take that out of the inclusion aspects.

Line 12-14: Suggest to change to, ‘The aim of this study was to evaluate, in a large group of dogs affected by lymphoma compared to dogs with other diseases, whether there is an alteration of hemostatic parameters in canine lymphoma and, if so, these were correlated with the type (B or T) of lymphoma and with disease stage.’

Line 58: Suggest to change to ‘…..stage of the disease (stage II to V).’

Line 68: Please change to ‘….cytopathological and histopathological evaluation of a lymph node, as well as flow cytometry and…’

Line 125: I wasn’t sure what this sentence was saying? ‘170 dogs with multicentric lymphoma fulfilled the more criteria selection.’

Line 128: Please change to, “Control-group resulted in 100% matched…’

Line 142-143: Please change to, ‘As shown, lymphoma dogs….and reduced Plt compared to sick (non-lymphoma) dogs.’

Line 145: Please change the title of Table 3 to ‘Comparison between lymphoma group and control group’

Figure 1: Please change the title to ‘Distribution of platelets between groups and lymphoma subgroups’ and please call the groups ‘Lymphoma B’ or ‘B Lymphoma’ and ‘Lymphoma T’ or ‘T lymphoma’.

Line 173: Please change to, ‘…can influence the coagulation system…’

Line 178: Please change to ‘….systems are hypothesized…’

Line 180-181: I think the false positive results are confusing for readers not as interested in coagulation so would recommend to elaborate on this point although I do not think this is as relevant.

Line 191: Please change to ‘….DIC that develop are a consequence…’

Line 193-195: This sentence is confusing to me as you did not assess for tissue factor in your study so I do not think you can say it was not observed in your study and I do not think this statement is relevant for the manuscript so would recommend to remove.

Line 209: There were no bone marrow results in the study so the thrombocytopenia result in stage V is very difficult to interpret.

Line 225: Please change to, ‘….display laboratory signs of possible hyperfibrinolysis.’

Comments on the Quality of English Language

I did not find this paper particularly interesting due to how subtle the changes were between the groups.  This is just coming from a coagulation perspective.

Author Response

We want to thank you because with your valuable suggestions and comments we hope to have improved our article. Thank you for allowing us to clarify some concepts.

Regarding the comment “If the electrophoresis and urinary parameters are not necessary to the authors to report, then I would take that out of the inclusion aspectsbiochemical-clinical analyzes are a routine part of the evaluations of patients with conditions, but our attention was focused on blood coagulation parameters. We have reformulated the sentence in the Materials and Methods in order to improve its interpretation. The staging procedures used in our study are all standardized as the analysis laboratory of the San Marco Clinic is a national reference diagnostic center for Veterinary Medicine.

Line 209: There were no bone marrow results in the study so the thrombocytopenia result in stage V is very difficult to interpret. We agree that the pathogenesis of every single hematologic complication is complex and more likely multifactorial and a full explanation was outside the descriptive scope of our work. We own a full detail of every case, but still, for example, a lack of megacariocytes in the bone marrow would not explain, in a single case, the thrombocytopenia, since we could have, simultaneously, the trapping of platelets in the spleen and/or an immune-mediated peripheral thrombocytopenia. We tried to briefly share the complexity of this topic with the readers.

12.20.2023

Based on the indications provided by the editor regarding the length of the text, we have made the necessary corrections. It follows that some parts of the text and in particular the “Discussion”, highlighted in yellow, have undergone a remodelling. We apologize for the confusion caused.

FQ

Reviewer 3 Report

Comments and Suggestions for Authors

This manuscript aims to evaluate whether there is an alteration of hemostatic parameters in canine multicentric lymphoma. In my view, the article submitted approaches the issue in an appropriate manner, but there are some minor suggestions.

Title: I propose to change to a title that specifies and emphasizes the strength of this study.

Animals and method: Detailed information is needed about diagnosis of lymphoma. Who performed cytological and histopathological evaluation? What devices were used to perform cytometry? What about bone marrow sampling method and analysis? How did you evaluate the spleen or liver?

Blood collection and sample analysis: When was blood collected? Were all samples collected before treatment in lymphoma and control groups?

Line 68: Change ‘cytophatological’ to ‘cytopathological’.

Line 125: What does ‘fulfilled the more criteria selection’ mean?

Line 107, Line 128: It seems that the order of Table 1 and Table 2 was switched.

Line 130: I suggest convert the units to ‘years’ instead of ‘months’.

Line 131: Delete the ‘kg’ in parentheses.

Line 131: I suggest convert the units to ‘years’ instead of ‘months’. Also, move the units after the parentheses.

Line 143: I recommend change ‘sick dogs’ to ‘unaffected dogs’.

Line 150: State that ‘platelet’ did not show statistically significant difference.

Line 152: I recommend change ‘sick dogs’ to ‘control group’.

Line 182: How about D-dimer?

Line 195: Please describe the reason why you confirm that the increase in TF was not identified in this study (including the result of prothrombin time).

Discussion:

1)     You suggested increased consumption, pathological fibrinolysis, and HPS as the causes of lower fibrinogen and platelet in stage V lymphoma. However, no difference in other variables was identified (e.g. D-dimer). I recommend you to discuss this with some recent references (Boyé P et al., (2021). Prognostic value of pretreatment plasma D-dimer level in dogs with intermediate to high-grade non-Hodgkin lymphoma. Vet Comp Oncol. 19(1):44-52.; Han HJ and Kim JH (2022). Correlation between d-dimer concentrations and thromboelastography in dogs with critical illness. Front Vet Sci. 844022.)

2)     Patients with diseases other than lymphoma were included in the control group, and how can the effects of the diseases be interpreted? Please make a mention about this in discussion part. Are there any tumor patients in the control group?

Comments on the Quality of English Language

Minor editing of English language required.

Author Response

We want to thank you because with your valuable suggestions and comments we hope to have improved our article.

Following your instructions, we changed the title of our work with the hope of making it more appropriate to the topic.

We have also reversed the order of Tables 1 and 2 as you indicated.

Furthermore, we have also included the bibliographical references recommended by you in the discussion (1) and, regarding point 2, The patients in the control group all had diseases for which the involvement of coagulation factors would not have been expected. None of the animals in this group, as reported in Materials and Methods, presented macroscopic neoplastic findings.

Title: I propose to change to a title that specifies and emphasizes the strength of this study. We could propose: “Evaluation of hemostatic dysregulation in canine multicentric lymphoma”

Animals and method: Detailed information is needed about diagnosis of lymphoma. Who performed cytological and histopathological evaluation? What devices were used to perform cytometry? Cytomics FC 500 Beckman Coulter cytofluorometer. What about bone marrow sampling method and analysis? How did you evaluate the spleen or liver? Yes, with abdominal ultrasound.

Blood collection and sample analysis: When was blood collected? Were all samples collected before treatment in lymphoma and control groups? As reported in Materials and Methods, the blood sample was taken from the cephalic or jugular vein depending on the size of the dog before proceeding with any type of treatment.

20.12.2023

Based on the indications provided by the editor regarding the length of the text, we have made the necessary corrections. It follows that some parts of the text and in particular the “Discussion”, highlighted in yellow, have undergone a remodelling. We apologize for the confusion caused.

FQ
